# Monocytic Cell-Induced Phase Transformation of Circulating Lipid-Based Liquid Crystalline Nanosystems

**DOI:** 10.3390/ma13041013

**Published:** 2020-02-24

**Authors:** Angel Tan, Yuen Yi Lam, Xiaohan Sun, Ben Boyd

**Affiliations:** 1Drug Delivery, Disposition and Dynamics, Monash Institute of Pharmaceutical Sciences, Monash University, Parkville Campus, VIC 3052, Australia; yuen.lam1@monash.edu (Y.Y.L.); xiaohan.sun@monash.edu (X.S.); 2ARC Centre of Excellence in Convergent Bio-Nano Science and Technology, Monash University, Parkville, VIC 3052, Australia

**Keywords:** self-assembly, lipid-based liquid crystalline nanosystem, cell interaction, synchrotron small angle X-ray scattering, phase behaviour

## Abstract

Both lamellar and non-lamellar configurations are naturally present in bio-membranes, and the synthetic lipid-based liquid crystalline nano-assemblies, mimicking these unique structures, (including liposomes, cubosomes and hexosomes) are applicable in the controlled delivery of bioactives. However, it remains uncertain whether these nanosystems retain their original phase identity upon contact with blood circulating cells. This study highlights a novel biological cell flow-through approach at the synchrotron-based small angle X-ray scattering facility (bio-SAXS) to unravel their real-time phase evolution when incubated with human monocytic cells (THP-1) in suspension. Phytantriol-based cubosomes were identified to undergo monocytic cell-induced phase transformation from cubic to hexagonal phase periodicity. On the contrary, hexosomes exhibited time-dependent growth of a swollen hexagonal phase (i.e., larger lattice parameters) without displaying alternative phase characteristics. Similarly, liposomes remained undetectable for any newly evolved phase identity. Consequently, this novel in situ bio-SAXS study concept is valuable in delivering new important insights into the bio-fates of various lipid-based nanosystems under simulated human systemic conditions.

## 1. Introduction

The natural formation of lamellar and non-lamellar structures within living systems are essential for enabling vital biological functions. For example, mammalian cell and organelle membranes have been observed to transiently form the cubic periodicity for controlling osmolality, as well as the hexagonal phases for regulating the calcium pump within muscle cells [1,2,3,4]. Self-assembled liquid crystalline nanosystems, mimicking these unique, hierarchically ordered structures have been cleverly engineered using lyotropic lipid-based excipients to enable various biomedical applications. Examples include bio-imaging, bio-sensing, diagnostics, and the controlled delivery of bioactives [5,6,7,8,9]. The critical packing parameter (CPP) is a concept used to describe the likely impact of molecular geometry of lipid on self-assembled morphology [10]. Typically, amphiphilic molecules with a critical packing parameter (CPP) of > 1 (i.e., a molecular space, dominated by the large hydrophobic tail relative to the hydrophilic head) tend to form negatively curved, inverse phases in water. Some commonly known non-lamellar structures include the anisotropic reverse hexagonal (H_2_) phase, the isotropic bi-continuous inverse cubic (V_2_) phase, and the reverse micellar network (denoted L_2_) (Figure 1a). On the other hand, lipid molecules with CPPs of 0.5–1 produce positively curved, lamellar vesicles (LV, or more commonly known as liposomes) [11]. 

While liposomal formulations have long-found clinical use for therapeutic delivery [12], a few non-lamellar systems (i.e., the Camurus products) have recently reached regulatory approval for internal applications [13]. Successful commercialisation has driven much of the scientific attention towards understanding the biological behavior of these lipid-based liquid crystalline (LLC) systems in the bloodstream and in tissue cells. Small angle X-ray scattering (SAXS) analysis, based on the synchrotron source, is a powerful tool to ascertain the phase characteristics of various LLC nanosystems within seconds, as compared to the timeframe of hours on the conventional laboratory SAXS instrument [8,14,15,16]. It has enabled extensive investigations into the morphological stability and phase transition of LLC systems in water or phosphate buffer saline (PBS) across different environmental and biochemical parameters: Temperature [8,17], hydration state [17,18], ionic strength and biomacromolecule effects [15,19,20,21,22], as well as pH [23,24,25]. However, their phase behaviour under physiological systemic conditions have been underexplored, to date. A majority of their bio-interaction studies have placed emphasis on cytotoxicity, cellular uptake and intracellular distribution [11,26]. Only a few reports focusing on protein corona effects revealed that biological fluids (e.g., blood serum and artificial cell culture media) could alter the lattice parameters of the LLC systems [27,28], and possibly trigger a slow transition from one phase to the other [29]. Yet, it remains unclear how the presence of human immune cells encountered by these particles in circulation influence their structural qualities.

This work, thus, presents a novel study approach that combines biological cell flow-through system with the synchrotron-based small angle X-ray scattering analysis (bio-SAXS) to probe the effects of human blood circulating cells on the phase behaviour of different LLC nanosystems. Two non-lamellar systems (i.e., cubosomes and hexosomes) and a lamellar system (i.e., liposomes) were examined for their real-time phase events under dynamic interactions with human monocytic cells, THP-1 (which are known to be phagocytic according to the American Type Culture Collection, ATCC, commercial source). To exclude chemical degradation as a factor affecting the self-assembled structures, the ‘indigestible’ lipid phytantriol (PHY, Figure 1b), which lacks an esterase-susceptible ester group, was used as a parent lipid in all formulations. This bio-SAXS technique is demonstrated to support real-time assessment of the monocytic cell-induced changes in phase behaviour of the LLC nanosystems.

## 2. Materials and Methods

### 2.1. Preparation of LLC Nanosystems

The molecular structure of each formulation component is illustrated in Figure 1b. Cubosomes were prepared from phytantriol (PHY > 95%; DSM Nutritional Products, Kaiseraugst, Switzerland) and Poloxamer 388 (Synperonic F108; Sigma-Aldrich, Saint Louis, MO, USA) at a lipid-to-stabiliser mass ratio of 10:1. Hexosomes were prepared from PHY, vitamin E acetate (Vit-EA ≥ 96%, Sigma-Aldrich, Saint Louis, USA) and F108 at a mass ratio of 10:1:1. Liposomes were formed based on PHY and 1,2-distearoyl-*sn*-glycero-3-phosphoethanolamine-N-[amino(polyethylene glycol)-2000] (DSPE-PEG2000 > 99%; Avanti Polar Lipids Inc., Alabaster, AL, USA) at a molar ratio of 95:5. For each system, PHY was pre-dissolved in chloroform and vortex-mixed with the additive (i.e., Vit-EA in the case of hexosomes), prior to solvent evaporation under vacuum conditions (40 °C) overnight. To produce cubosomes and hexosomes, each thin film of lipids were dispersed in PBS containing F108, whereas liposomes were dispersed in PBS containing DSPE-PEG2000. Each dispersion was ultrasonicated (Misonix XL2000, Misonix Inc., Farmingdale, NY, USA) for 40 min in a pulse mode (i.e., 2 s on/off cycles) at an amplitude of 20%. Water was from a MilliQ system (Millipore, Sydney, Australia).

### 2.2. Particle Sizing and Cryogenic Transmission Electron Microscopy (Cryo-TEM) Analysis

The particle sizes of the nanosystems were analysed by dynamic light scattering (DLS), using a Malvern Zetasizer Nano ZS instrument for samples diluted to 100 μg/mL of PHY with water (at 25 °C). The cryo-TEM analysis was performed at the Bio21 Advanced Microscopy Facility (Parkville, Australia). Negatively stained grids were observed on a FEI Tecnai F30 instrument (Field Electron and Ion Company, Eindhoven, The Netherlands) operated at 300 kV, and equipped with a Gatan quantum 965 energy filter and an upper CETA 4k × 4k CMOS camera.

### 2.3. Biological Cell Flow-Through Setup for Small Angle X-Ray Scattering (Bio-SAXS) Analysis

Human-derived monocytic cells (THP-1, passage 22–24; ATCC^®^ TIB-202^TM^, Manassas, VA, USA) were cultured in a bio-incubator (37 °C, 5% CO_2_) at an initial density of 2 × 10^5^ viable cells/mL in a T75 flask, and sub-cultured when a density of 1 × 10^6^ cells/mL was reached. The cell cultures were replenished every 3 days with fresh RPMI-1640 medium supplemented with fetal bovine serum (FBS) at 10% (v/v) via centrifugation and removal of the old supernatant medium. Cells re-suspended at 20 × 10^6^ cells/mL were used for the experiments at the SAXS/WAXS beamline of the ANSTO Australian Synchrotron, Clayton, VIC, Australia [16].

In the bio-SAXS setup, an open-ended quartz capillary tube (Quartz 15-SQZ 1.5 mm, Charles Supper, Natick, MA, USA) was connected through the upper end (‘input’) to a circulating peristaltic pump (Ismatec^®^ ISM597D, Cole-Parmer, Wertheim, Germany) via silicone tubing, and the lower end (‘output’) opened to a glass vial reservoir to enable continuous recirculation of the medium. An aliquot of 0.5 mL cell suspension (20 × 10^6^ cells/mL) was circulated through the capillary at a flow rate of 20 mL/min (i.e., simulating the human arterial wall shear stress of 10 dynes/cm^2^) [26]. Each nanosystem (0.5 mL) was dispersed at 2 mg/mL in the RPMI-1640/FBS medium, and introduced into the cell-rich reservoir via a syringe and needle. This resulted in a total flow-through volume of 1 mL at a ratio of approximately 1 mg lipids: 1 × 10^7^ cells. The SAXS profiles were collected using a 13 keV X-ray at a camera-to-detector length of 1600 mm to acquire the scattering vector within a *q*-range of 0.01–0.6 Å^−1^ (calibrated against a silver behenate standard). The acquisition time was 10 s each for every 1 min.

## 3. Results and Discussion

### 3.1. Physicochemical Characteristics of Self-Assembled LLC Nanosystems

Figure 2a shows the cryo-TEM images of the negatively curved hexosomes (H_2_ phase), cubosomes (V_2_ phase), and the positively curved liposomes (LV). The H_2_ phase displayed the characteristic ring-like striations; the V_2_ phase was distinguished by the highly ordered, mesh-like network; and the LV system exhibited a mixture of uni- and multi-lamellar bilayer structures [14,30,31,32]. Particle sizing by DLS shows that the mean hydrodynamic diameters of the H_2_ and V_2_ particles span across the range of 250–400 nm (polydispersity index, PDI < 0.5), which are slightly larger than the LV system (<150 nm, PDI < 0.2). Nevertheless, these size data are comparable to other previous reports [33,34,35].

Phase characteristics at equilibrium were further established using the synchrotron-based SAXS technique. As opposed to the existing SAXS profiling of the LLC dispersed systems that typically employed lipid concentrations at > 10 mg/mL, we attempted to identify the lowest possible lipid concentrations that meet both, the SAXS instrument detection limits and the biocompatibility levels (i.e., < 1 mg/mL per 10^6^ cells), according to previous cytotoxicity studies across different cell lines [11]. As shown in Figure 2b, the LLC phase-specific diffraction peaks were reliably acquired for each system at 1 mg/mL lipid level (i.e., ideal for 10^7^ cells in the subsequent studies). The H_2_ phase (primary peak at *q* = 0.141 Å^−1^) shows three Bragg peaks at a spacing ratio of 1:√3:√4. The V_2_ phase was identified to be of a double diamond (Pn3m) geometry with a peak spacing ratio of √2:√3:√4:√6 (primary peak at *q* = 0.129 Å^−1^). On the other hand, the vesicular structure (LV) is characterised by a broad, diffuse band, as shown previously by other researchers [24,30,36]. The presence of THP-1 cells was also confirmed to exert negligible interference on the diffraction peak detection.

### 3.2. Phase Transformation of LLC Systems was Triggered by Monocytic Cells but not by Cell Culture Medium

We coupled the biological cell flow-through a capillary system with the synchrotron-SAXS to enable instantaneous acquisition of the diffraction patterns for each nanosystem every 1 min, under dynamic bio-interaction conditions. It is interesting to observe that the three different types of LLC nanosystems exhibited dissimilar phase events during their encounter with monocytic cells.

Hexosomes retained the pure H_2_ phase after incubation in the cell-free, FBS-containing medium for 30 min (Figure 3, control reference). The medium caused a slight shrinkage of the internal structure based on the primary peak shift from *q* = 0.141 Å^−1^ (Figure 2b) to 0.147 Å^−1^ (Figure 3). When monocytic cells were introduced into the circulation, new diffraction peaks initiated at the *q*-position of 0.123 Å^−1^ started to appear within 8 min (Figure 3, treatment). The intensity of the new diffraction peaks increased continuously while the original peak signals subsided over the next 24 min. Correlation with the Miller indices for known phases indicates the identity of H_2_ phases for both the original and the evolved structures (i.e., 1:√3:√4).

Similarly, cubosomes maintained the pure V_2_ phase in the cell-free medium for 30 min (Figure 4, control reference). However, the presence of monocytic cells triggered a phase transformation from a V_2_ phase to a mixed V_2_/H_2_ phase from 8 min onwards, where the peak intensity of the evolved H_2_ phase (primary *q* = 0.126 Å^−1^) increased progressively until the V_2_ phase became almost undetectable at the end of the 24 min analysis (Figure 4, treatment). The complete V_2_-to-H_2_ phase transformation, which could be interpreted as cell-triggered “instability”, is somewhat in agreement with other findings showing the higher fusogenic propensity of PHY-based cubic systems with cellular membranes in comparison with the hexagonal equivalents [11].

The mean lattice parameter, *a*, was deduced from the scattering vector (*q*) and the inter-planar spacing (*d*) of each unit cell according to equation (1) for H_2_ phase, and equation (2) for V_2_ phase, respectively,
*a_hexagonal_* = 4*d*/3(*h*^2^ + *k*^2^)^0.5^(1)
*a_cubic_* = *d*(*h*^2^ + *k*^2^ + *l*^2^)^0.5^(2)
where *d* = 2π/*q*, and the values *h*, *k* and *l* are the Miller indices of the Bragg peaks [37].

From the trend of lattice parameter (Figure 5), it could be seen that the evolution of a swollen H_2_ phase in both hexosomes and cubosomes coincided within the same range of mean lattice parameter (i.e., 57 Å ≤ *a* ≤ 61 Å). The appearance of the cell-triggered H_2_ periodicity is intriguing and remains to be further elucidated, however the transfer of lipids between particles is known to take place, driven by entropy of mixing [38,39]. The transfer of lipid out of cubosomes would not induce a change in lattice parameter of the cubic phase, yet the gradual shift in lattice parameter, and ultimate transition to hexosomes is strongly suggestive of transfer or extraction of lipids from the cells into the cubosomes. The effect is not due to proteins in the media because the control particles in media did not exhibit a change in structure. Likewise, the hexagonal phase particles are transitioning slowly to lower lattice parameter following the behavior of the cubosomes which also suggested transfer of lipids from the cells into the particles, in order to induce such a change. The aforementioned mechanism does not preclude a fusogenic event, which may be the trigger for the change in particle structure, but the data certainly does not suggest attrition of lipid from the host particles into the cells through solution, as it is believed to be the case in a mixture of cubosomes with emulsions [38].

In the case of liposomes, the typical broad diffusive peak pattern was present throughout the 20 min of contact with monocytic cells (Figure 6). There was no detection of any identifiable new phase characteristics. This possibly reflects relatively good stability of the liposomal system as conferred by the traditional PEGylation “stealth” approach. Interfacial functionalisation with PEG has long been reported in minimising non-specific interactions with serum proteins and systemic clearance by the phagocytic cells [40,41]. However, the absence of detection may also be attributed to the limit of SAXS detection sensitivity. Nevertheless, the current bio-SAXS setup has successfully facilitated real-time determination of the bio-fate of three different LLC nanosystems in a simplified systemic environment.

## 4. Conclusions

This work demonstrates the viability and importance of coupling biological cell flow-through circuits and the synchrotron-based SAXS analysis (bio-SAXS) to understand the systemic bio-fate of various LLC nanostructured systems. It has generally been assumed that these thermodynamically stable nanosystems retain their original identities when applied as injectable or implantable drug delivery systems. The current findings highlight that the studied non-lamellar systems (i.e., cubosomes and hexosomes stabilised using the traditional Poloxamer-based agents were prone to modifications of their internal nanostructure (i.e., time-dependent loss of original phase characteristics) in the presence of blood circulating immune cells. In contrast, the equivalent liposomal system stabilised by PEG did not show any signs of instability upon cellular contact. Therefore, behaviour of such systems in applications such as drug release or interfacial interactions examined under cell-rich environments needs to be interpreted with care. This bio-SAXS experimental configuration will be applicable to support the process of formulation optimisation and to predict the systemic behaviour of various LLC-based drug delivery systems under cellular-rich physiological environments.

## Figures and Tables

**Figure 1 materials-13-01013-f001:**
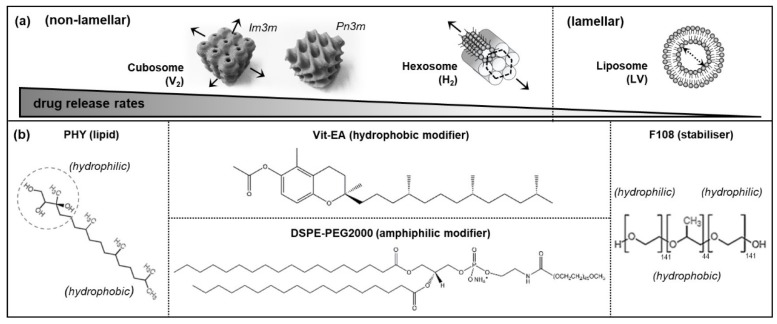
(**a**) Schematics depicting the non-lamellar and lamellar structures and their corresponding drug release rates, where the arrows indicate the possible paths of cargo release; (**b**) molecular structure of the formulation components: the parent lipid, phytantriol (PHY); hydrophobic modifier, Vitamin E acetate (Vit-EA); amphiphilic modifier, 1,2-distearoyl-*sn*-glycero-3-phosphoethanolamine-N-[amino(polyethylene glycol)-2000 (DSPE-PEG2000); and stabiliser, Poloxamer 388 (F108).

**Figure 2 materials-13-01013-f002:**
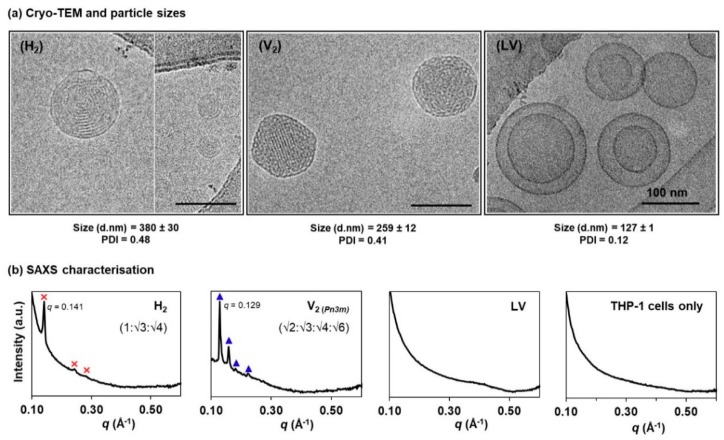
Structural characterisation of the lipid-based liquid crystalline (LLC) nanosystems: (**a**) Cryogenic transmission electron microscopy (cryo-TEM) images, and the corresponding hydrodynamic diameters (d. nm) and polydispersity index (PDI) determined using dynamic light scattering (DLS) technique; (**b**) synchrotron-based small angle X-ray scattering (SAXS) characterisation of the phytantriol-based hexosomes [H_2_, where the Bragg peaks, ✕, could be indexed as Miller indices (*hk*) = (10), (11) and (20)], cubosomes [V_2_, where the peaks, ▲, corresponded to Miller indices (*hkl*) = (110), (111), (200) and (211]), and liposomes (LV) at 1 mg/mL lipid level, as well as the blank THP-1 cells suspended in cell culture medium (i.e., 1 × 10^7^ cells in RPMI-1640 supplemented with 10% FBS).

**Figure 3 materials-13-01013-f003:**
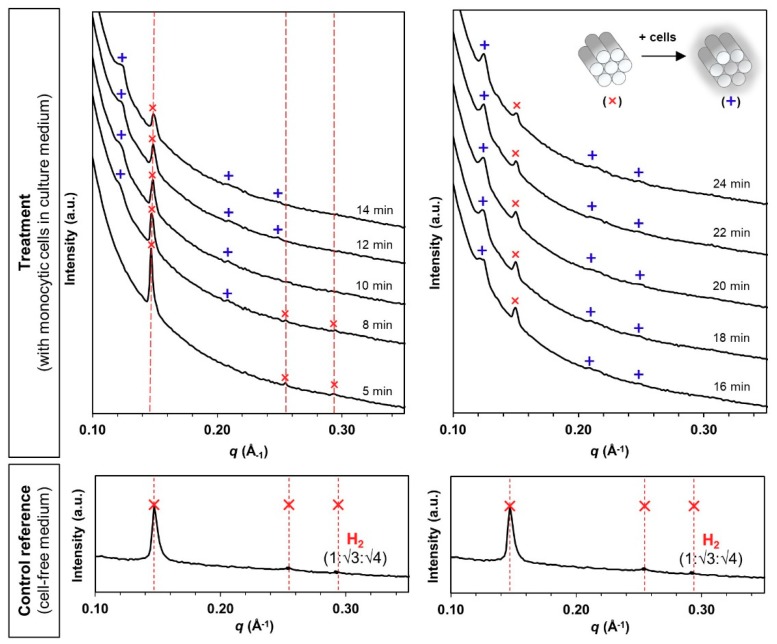
Bio-SAXS profiles of the phytantriol-based hexosomes (H_2_ phase) at 1 mg/mL under circulating flow conditions in the absence and presence of human monocytic cells (at 27 °C hutch temperature): original H_2_ phase (✕, primary *q* = 0.147 Å^−1^) and evolved H_2_ phase (+, primary *q* = 0.123 Å^−1^) at Bragg peak spacing ratios of 1:√3:√4 [Miller indices (*hk*) = (10), (11) and (20)].

**Figure 4 materials-13-01013-f004:**
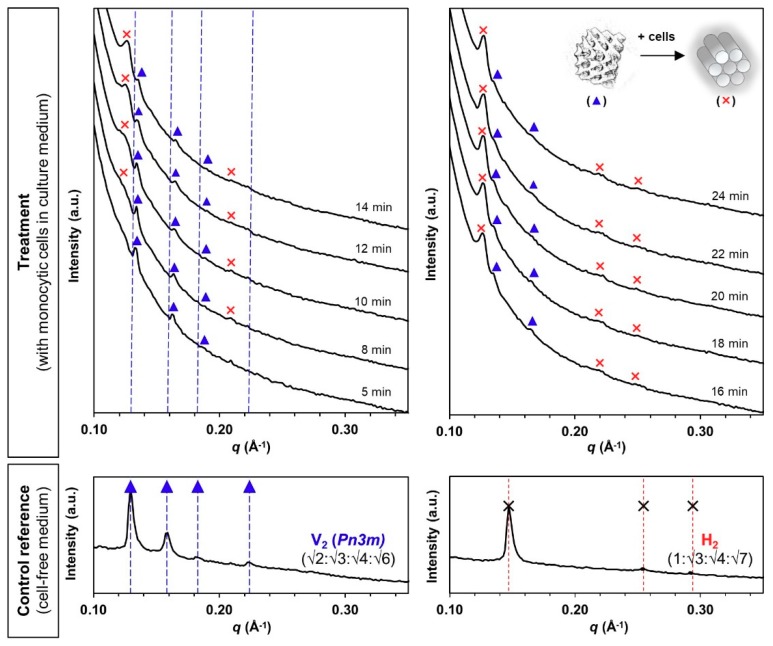
Bio-SAXS profiles of the phytantriol-based cubosomes (V_2_ phase) at 1 mg/mL lipid level under circulating flow conditions in the absence and presence of human monocytic cells (27 °C): original V_2_ phase (▲, primary *q* = 0.129 Å^−1^) at a Bragg peak spacing ratio of √2:√3:√4:√6 [Miller indices (*hkl*) = (110), (111), (200) and (211)], and evolved H_2_ phase (✕, primary *q* = 0.126 Å^−1^) at a Bragg peak spacing ratio of 1:√3:√4 [Miller indices (*hk*) = (10), (11) and (20)].

**Figure 5 materials-13-01013-f005:**
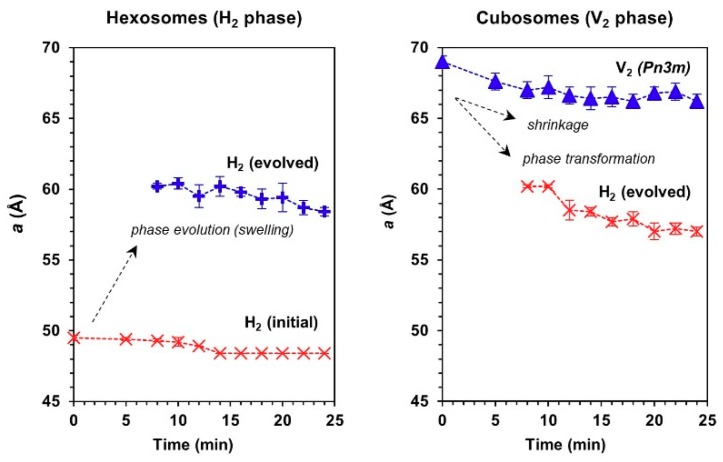
Changes in the mean lattice parameter, *a*, of hexosomes (left panel) and cubosomes (right panel) in contact with human monocytic cells under circulating flow conditions (27 °C).

**Figure 6 materials-13-01013-f006:**
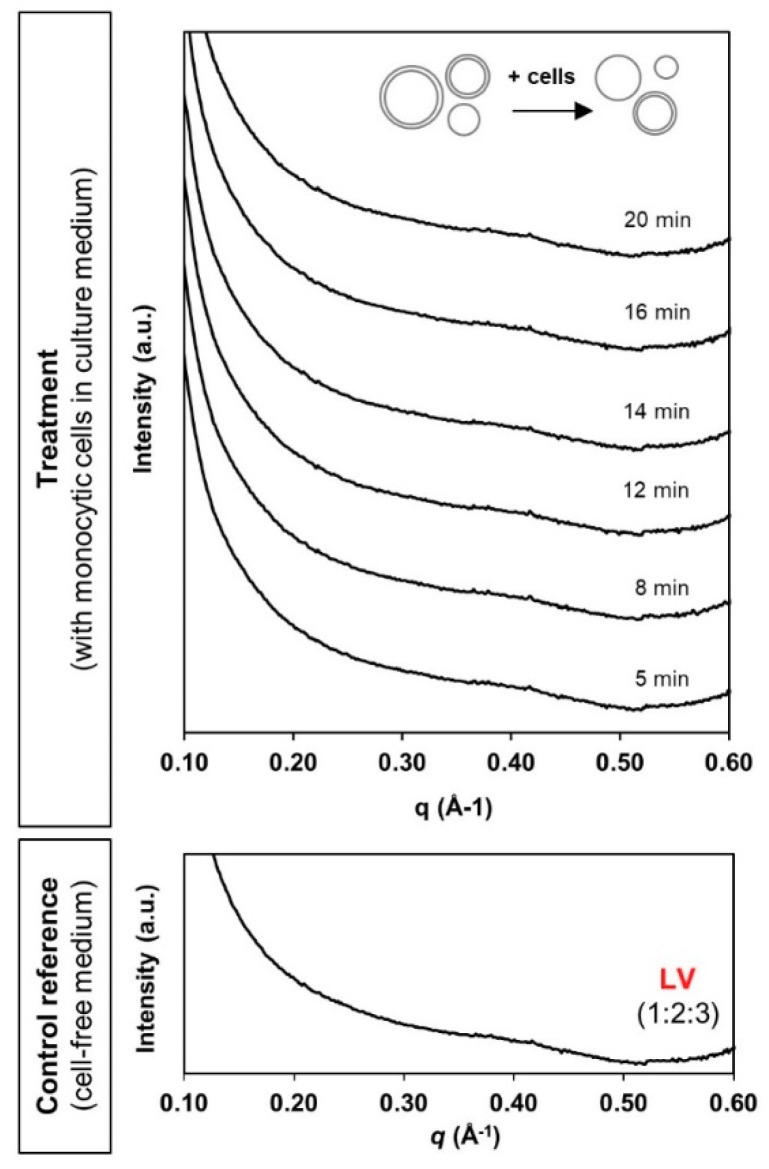
Bio-SAXS profiles of the phytantriol-based liposomes (LV) at 1 mg/mL lipid level under circulating flow conditions in the absence and presence of human monocytic cells (27 °C).

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
