# Peer review of "Monocytic Cell-Induced Phase Transformation of Circulating Lipid-Based Liquid Crystalline Nanosystems"

_materials, 2020, doi:10.3390/ma13041013_

Round 1

Reviewer 1 Report

Compliments

Author Response

We appreciated the complimentary remarks from Reviewer 1.

Reviewer 2 Report

I have read the manuscript entitled, 'Monocytic cell-induced phase transformation of circulating lipid-based liquid crystalline nanosystems' with high interest. The concept of the article is novel and the results are well supported by the experiments. Hence I am recommending to publish this manuscript as its present form.

Author Response

We appreciated the complimentary comments from Reviewer 2.

Reviewer 3 Report

The manuscript describes the liquid crystalline phase transformations (e.g. from cubic to hexagonal) induced by blood circulating cells. The real-time phase evolution was studied by in situ synchrotron-based SAXS (i.e. bio-SAXS) measurements. This is an overall interesting and multidisciplinary article that demonstrates the potential bio-applications of liquid crystals.

The paper could be suitable for publication in "Materials" only after two minor considerations:

1) It would be helpful for readers to include the Miller indeces in the SAXS patterns (Figure 2b, 3 and 4)

2) A revision of the references should be made in order to explain current interest in: liquid crystals for bioapplications [Nature Mater. 2007, 6, 929; Curr. Appl. Phys. 2012, 12, 1387‐1412; Liquid Crystals in Living Systems and Liquid Crystals in the Development of Life. In Handbook of Liquid Crystals (2014, eds J.W. Goodby, C. Tschierske, P. Raynes, H. Gleeson, T. Kato and P.J. Collings)], self-assembled systems studied by SAXS (Macromolecules 2016, 49, 7825-7836; Angew. Chem. Int. Ed. 2013,
52, 8934-8937)
